# Automated Rice Phenology Stage Mapping Using UAV Images and Deep Learning



Xiangyu Lu [1,2], Jun Zhou [1,3], Rui Yang [1], Zhiyan Yan [4], Yiyuan Lin [1], Jie Jiao [1] and Fei Liu [1,2,*]

1 College of Biosystems Engineering and Food Science, Zhejiang University, Hangzhou 310058, China
2 Huanan Industrial Technology Research Institute, Zhejiang University, Guangzhou 510700, China
3 College of Mechanical and Electrical Engineering, Xinjiang Agricultural University, Urumqi 830052, China
4 Institute of Agricultural Economics and information, Jiangxi Academy of Agricultural Sciences, Nanchang 330200, China
* Correspondence: fliu@zju.edu.cn; Tel.: +86-571-8898-2825

**Abstract:** Accurate monitoring of rice phenology is critical for crop management, cultivars breeding, and yield estimating. Previously, research for phenology detection relied on time-series data and orthomosaic and manually plotted regions, which are difficult to automate. This study presented a novel approach for extracting and mapping phenological traits directly from the unmanned aerial vehicle (UAV) photograph sequence. First, a multi-stage rice field segmentation dataset containing four growth stages and 2600 images, namely PaddySeg, was built. Moreover, an efficient Ghost Bilateral Network (GBiNet) was proposed to generate trait masks. To locate the trait of each pixel, we introduced direct geo-locating (DGL) and incremental sparse sampling (ISS) techniques to eliminate redundant computation. According to the results on PaddySeg, the proposed GBiNet with 91.50% mean-Intersection-over-Union (mIoU) and 41 frames-per-second (FPS) speed outperformed the baseline model (90.95%, 36 FPS), while the fastest GBiNet_t reached 62 FPS which was 1.7 times faster than the baseline model, BiSeNetV2. Additionally, the measured average DGL deviation was less than 1% of the relative height. Finally, the mapping of rice phenology was achieved by interpolation on trait value–location pairs. The proposed approach demonstrated great potential for automatic rice phenology stage surveying and mapping.

**Keywords:** rice phenology; image segmentation; deep learning; UAV images; direct geo-locating

## 1. Introduction

Rice cultivation has a long history, and it is one of the world's main cereal crops, as well as a primary source of food for more than three billion people worldwide [1]. The requirements of rice vary according to its growth stage, and different management techniques are required at various stages, such as drying the fields at the late tillering stage to elongate plant stems [2]. Moreover, for rice breeding, the importance and reliability of key traits vary from stage to stage. For example, the rice leaf area index (LAI) should be estimated after the jointing stage [3], while yield should be estimated at the flowering stage or later [4,5]. The phenology in different seasons and the duration of each stage will influence the growth of rice as well [6,7]. Therefore, to cultivate rice well, monitoring the rice growing stage is important, especially for field planting.

Field inspections by workers, even with the aid of a detection model using handheld camera images, are not practical for large areas [8]. A growth simulation model based on environmental parameters such as thermal time accumulation is not stable and accurate [9]. With the trend towards intensive farmland management, remote sensing using satellites or drones offers a highly efficient method of investigation. Satellites can observe a larger area at the scale of a county or province [10]. Moreover, a synthetic aperture radar (SAR) is used for estimating the historical rice growing and planting pattern [11]. Most of these methods rely on time-series data to identify temporal responses of the entire season, which are

hard to realize in real-time surveying [12]. Additionally, the temporal–spatial resolution of satellites (greater than 10 m and five days) is too coarse for use in precision agriculture [13].

Drones are widely used in plant phenotype, including crop identification [14], crop biomass estimation [15,16], nutrition assessment [17], seedling counts [18], and also phenology investigation. Compared with infrared imaging capturing the thermal irradiance of the target [19], high-resolution RGB and spectral imaging are more effective for observing detailed rice phenological traits. Yang et al. [2] constructed digital surface models (DSMs) based on UAV images to estimate rice plant height and the growth stage, which could help guide the automated irrigation system for water-saving. Various features, including the vegetation index, color space, and textures, were extracted from regions of interest (ROI) in orthomosaic maps [20]. These features were subsequently processed using ensemble machine-learning algorithms for phenology detection. Ma et al. [21] utilized the red-edge chlorophyll index (CIred edge) and normalized difference vegetation index (NDVI) to monitor the hybrid rice's initial heading stage. The results verified that the CIred edge is more suitable for monitoring the phenological stage. Additionally, there have been numerous studies that have utilized UAV images to quantify important rice growth characteristics, such as the plant area index (PAI) [22], leaf dry biomass (LDB) [23], and storage organ biomass [24]. However, many of these studies have relied on time-series (multi-temporal) vegetation index (VI) data for the rice growth stage, which are not robust for expansion or generalization. Yang et al. [25] proposed a convolutional neural network (CNN) that utilizes mono-temporal UAV images for detecting rice phenological stages and received an accuracy of 83.9% with an auxiliary regional mean thermal time input. Despite its promise, this classification model relies on manually plotted ROIs and requires the generation of orthomosaic maps before phenology detection.

Deep learning has demonstrated superior performance in pattern recognition [26–28], and the multi-class image segmentation model has the potential for fully automatic target extraction and mapping [29]. Researchers Lan et al. [8], Deng et al. [30], and Sai et al. [31] have developed several semantic segmentation networks for real-time identification and mapping of weeds in paddy fields using UAV. In [32], a paddy field segmentation model was proposed that combines an attention mechanism with an adaptive spatial feature fusion algorithm based on DeepLabv3+. This proposed model, referred to as SA-DeepLabv3+, attained a higher accuracy and speed. To identify the rice lodging area, Yang et al. [33] used an FCN-AlexNet network and added an extra vegetation index map to the input image to identify areas of rice lodging. However, there have been few studies on rice field area segmentation and multi-stage discrimination.

To address these challenges, the objectives of this study are (1) to create a dataset for rice field segmentation and phenology classification; (2) to develop a fast and accurate model for multi-class segmentation and compare it to existing models; (3) to investigate an effective and reliable workflow for locating and mapping traits using UAV images sequence; (4) to test the effectiveness of the proposed system with a rice phenology mapping test.

## 2. Materials and Methods

### 2.1. Experimental Sites and Equipment

To ensure data diversity, unmanned aerial vehicle (UAV) images of rice fields were collected from 14 flight campaigns in the main producing provinces of Zhejiang, China, from June 2021 to September 2022. These campaigns were conducted at 9 different experimental fields, including djd1~djd5, qt1, qt2, sds, xs2 in Hangzhou and lq1, lq2, ra1, ra2 in Wenzhou City, as shown in Figure 1. On the same day of the UAV flight, rice planting experts investigated and recorded the ground truth stage in the field.

All aerial images were collected by a Zenmuse P1 (DJI Technology Co., Ltd., Shenzhen, China) full-frame camera mounted on the Matrice 300 RTK quadcopter (DJI Technology Co., Ltd., Shenzhen, China) through a three-axis gimbal at the vertical overhead view. The camera was equipped with a 35 mm focal length lens, and its sensor size, photo size, and pixel size are 35.9 × 24 mm, 8192 × 5460, and 4.4 μm, respectively [34]. Other

equipment includes an i70II handheld RTK (CHCNAV, Shanghai, China) for ground control point (GCP) locating, a server with two Nvidia RTK-2080Ti GPUs for deep learning model training, and a computer with Nvidia RTX-2070 GPU for model speed test and other data processing.

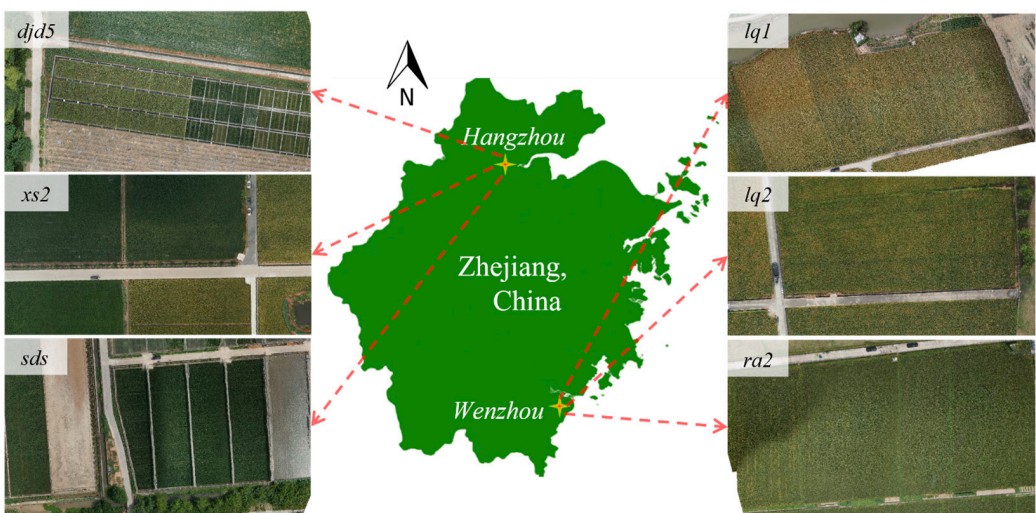

**Figure 1.** Experimental Sites Location and Orthophotos of Several Sites.

As for the software environment, Pix4D Mapper 4.4.12 (Pix4D SA, Prilly, Switzerland) software was used to generate orthomosaics through SFM and calibrate camera parameters [35]. QGIS 3.24 (OSGeo, Chicago, IL, USA) software was used to visualize the waypoints and orthophotos for each flight and also for interpolating and mapping the distribution of the trait [36]. To annotate segmentation mask labels effectively, we adopted the interactive annotation software EISeg (Baidu, Beijing, China) [37]. Additionally, the construction of deep learning models was based on the PyTorch framework and MMSegmentation toolbox [38], and other data processing and algorithm were implemented with Python 3.8 scripts.

*2.2. Construction Process of Dataset*

To provide a reliable data basis for rice filed segmentation and phenology classification of UAV images, we used this section to illustrate the flow and techniques of building the Paddy Segmentation dataset (PaddySeg), as shown in Figure 2.

All UAV flights were at 25 m relative height above ground with the camera lens vertical to the ground, such that the ground sampling distance (GSD) of the original P1 image was around 0.31 cm/pixel. During each flight, the onboard real-time kinematic system (RTK) was used to acquire high positioning accuracy of horizontal 1 cm [39]. The forward and side overlap ratios were both 70% to ensure abundant feature matching at the structure-from-motion (SFM) process, and the average flight speed was 2.5 m/s. There are no strict limitations on weather or illumination, and the camera was set to P mode of automatic exposure to facilitate the diversity of the image data and robustness in subsequent model training. The waypoints and routes were planned automatically after selecting the target field. Moreover, all images were retrieved back from the SD card onboard after the flight campaign.

In most cases, the original size of the aerial image (8192 × 5460) is too large to process in deep learning models, causing out-of-memory in training and high latency during testing. Moreover, not all pixels in the image are worth calculating, considering the high resemblance in the farmland scene and the high overlap of aerial images. Moreover, smaller patches of the image are more convenient and fluent at labeling.

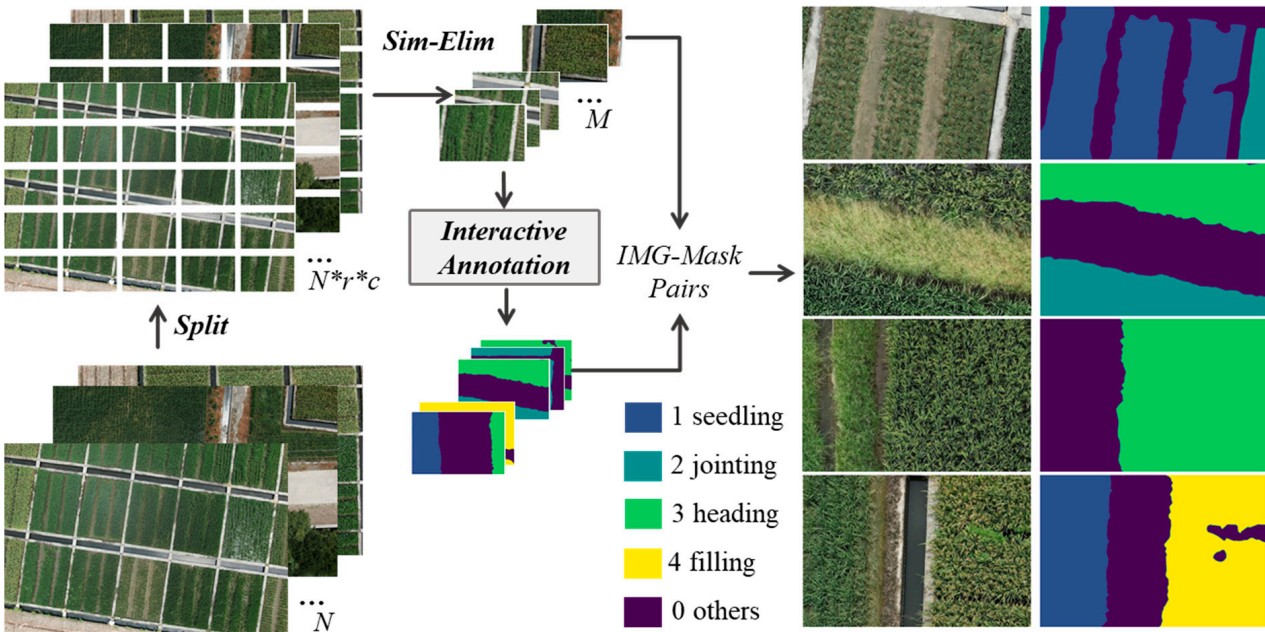

**Figure 2.** The construction process of PaddySeg dataset.

Consequently, each image was first divided into 5 rows and 5 columns of patches of 1638 × 1092 pixels (around 5.1 m × 3.4 m field of view), and patches were selected manually to eliminate similar (Sim-Elim) or redundant features. For example, patches containing more than one period of rice and other plants or weeds are preferred, while patches full of the crop are undesirable. After this process, the number of patches was reduced from $N * r * c$ to $M$, where $N$ is the original image number of one mission and $M$ is the target patches number according to the data diversity of the mission.

According to the management demands and key feature differences, we divided the rice phenology into 4 stages: 1. *seedling*: from seedling to the tillering stage, around 25 days after transplanting; 2. *jointing*: from jointing to booting stage, around 30 days; 3. *heading*: from heading to the flowering stage; and 4. *filling*: from filling to ripening stage. Each stage has its unique phenotype. At the seedling, bare land can be observed between the rice plants of one row, while this window is closed at the jointing stage when the grown-ups cover the land. Thanks to the high resolution of the P1 camera, we could observe the rice flowers in the photo of the heading stage and also the rice ear in the filling stage.

As the right part of Figure 2 shows, apart from the rice region, other vegetation, bare land, field ridges, or weeds were considered as "others", which is labeled as 0. We used EISeg to label the foreground paddy region and phenology stage by adding positive and negative sample points and generating corresponding masks. Table 1 shows the sample number, class pixel number, and distribution across the different missions. Although some of the campaigns have only one plot of rice (and only one stage in the mission), others contain multi-stages, especially for djd and qt sites where rice seedlings were transplanted at different seasons. It is worth noting that the total pixel numbers of each class are imbalanced, which may lead to majority bias during training [40], and we will discuss this later.

After gathering all 2600 pairs of image-mask, we split the dataset into 3 parts randomly with a constant seed 2022. The sample number and ratio of training, testing, and validation are 1820, 520, 260, and 7 : 2 : 1.

**Table 1.** Samples Number and Stages Distribution.

| Mission Code | Patch Number | Class Pixel Number (Million) | | | | |
|---|---|---|---|---|---|---|
| | | Seedling | Jointing | Heading | Filling | Others |
| 210604_djd1 | 200 | 208.8 | 0 | 0 | 0 | 149.1 |
| 210606_djd2 | 140 | 162.0 | 0 | 0 | 0 | 88.5 |
| 210909_djd3 | 200 | 0 | 244.4 | 0 | 0 | 113.4 |
| 210616_djd4 | 200 | 0 | 259.5 | 0 | 0 | 98.4 |
| 210718_djd5 | 300 | 34.3 | 169.0 | 0 | 186.2 | 147.3 |
| 210721_djd6 | 300 | 55.0 | 184.5 | 0 | 146.4 | 150.8 |
| 220628_qt1 | 300 | 87.3 | 6.5 | 281.4 | 2.5 | 159.0 |
| 220628_qt2 | 260 | 76.0 | 73.4 | 140.9 | 0 | 174.9 |
| 220712_lq1 | 100 | 0 | 0 | 2.0 | 117.7 | 59.2 |
| 220712_lq2 | 100 | 0 | 0 | 0 | 119.4 | 59.5 |
| 220713_ra1 | 100 | 0 | 0 | 0 | 119.1 | 59.8 |
| 220713_ra2 | 100 | 0 | 0 | 0 | 122.7 | 56.2 |
| 220727_sds | 140 | 0 | 168.6 | 0 | 0 | 81.9 |
| 220928_xs2 | 160 | 0 | 2.8 | 176.5 | 6.8 | 100.2 |
| Sum | 2600 | 623.4 | 1108.7 | 600.8 | 820.8 | 1498.2 |
| Pixel Ratio | - | 13% | 24% | 13% | 18% | 32% |

## 2.3. Multi-stage Rice Field Segmentation Model

Several studies and experiments demonstrated the efficiency of bilateral structure in semantic segmentation tasks [8,41]. With a detailed branch collecting low-level spatial details and a semantic branch extracting high-level semantics, this kind of model greatly accelerates the inference while improving the segmentation performance [41]. Due to the amount of aerial data and real-time inferencing demands, a faster model is required. However, the original detailed branch and decoder head of BiSeNetV2 demand heavy computational expenses of nearly 90%, as shown in Table 2 [42].

**Table 2.** Computation and Memory consumption of BiSeNetV2 [42].

| Computation/ Parameters | Overall | Components | | | | |
|---|---|---|---|---|---|---|
| | | Detail Branch | Semantic Branch | BGA Layer | Decode Head | Auxiliary Head |
| FLOPs (G) | 21.29 | 10.11 | 1.22 | 1.53 | 8.43 | 0.00 |
| | 100.0 | 47.5 | 5.7 | 7.2 | 39.6 | / |
| Weights (M) | 3.34 [1] | 0.52 | 1.16 | 0.48 | 1.19 | 11.42 |
| | 100.0 | 15.5 | 34.7 | 14.3 | 35.4 | / |

[1] Auxiliary Head weight is not included in the total for the inferencing process.

Consequently, we implement an efficient ghost convolution module and construct the Ghost Bilateral Network (GBiNet) by improving the computational bottlenecks in BiSeNetV2. In Section 2.3.1, the ghost convolution module and other basic components are introduced. Section 2.3.2 describes the overall structure of GBiNet, and other experimental settings and parameters are detailed in Section 2.3.3.

### 2.3.1. Ghost Convolution Module (GCM)

There are many similar and redundant feature maps in a well-trained convolutional neural network to ensure a comprehensive and stable understanding of the image data. As shown in Figure 3, after the first stage of the Detail Branch in BiSeNetV2, the output feature maps contain many similar pairs or groups. In each group, one feature map may convert to another through a cost-effective linear transformation [43].

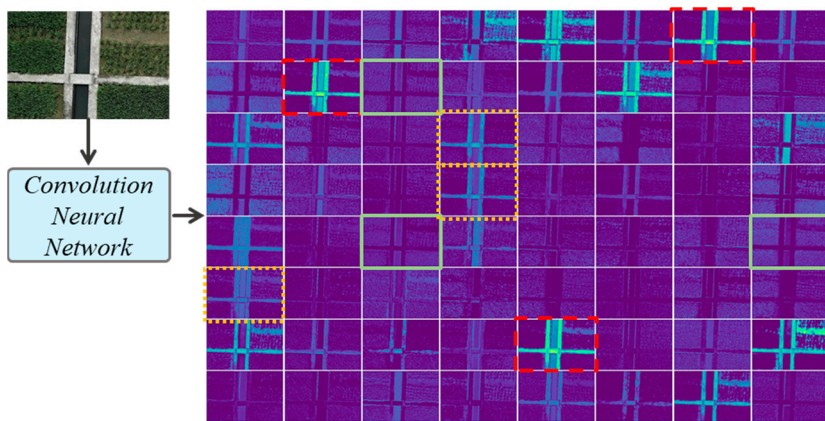

**Figure 3.** Similar Feature Maps in Convolution Neural Networks.

Figure 4a illustrates the process of ghost convolution, and it can be denoted as:

$$Y_0 = Conv_{3\times3}(X, s, c'), \tag{1}$$

$$Y = Concat([Y_0; G_{Linear}(Y_0, r)]) \tag{2}$$

where $X \in \mathbb{R}^{c \times h \times w}$ are the input feature map of $c$ channels and $h \times w$ size, $Conv_{3\times3}$ is the vanilla convolution with $3 \times 3$ kernel size, $s$ stride, and $c'$ kernels, $Y_0 \in \mathbb{R}^{c' \times \frac{h}{s} \times \frac{w}{s}}$ is the initial feature map of $c'$ channels and $\frac{h'}{s} \times \frac{w'}{s}$ size, and $G_{linear}$ is the group linear transformation that generates $(r-1)$ ghosts feature maps from each channel of $Y_0$, and $r \in N^*$ is the ratio of ghost feature maps at output $Y$.

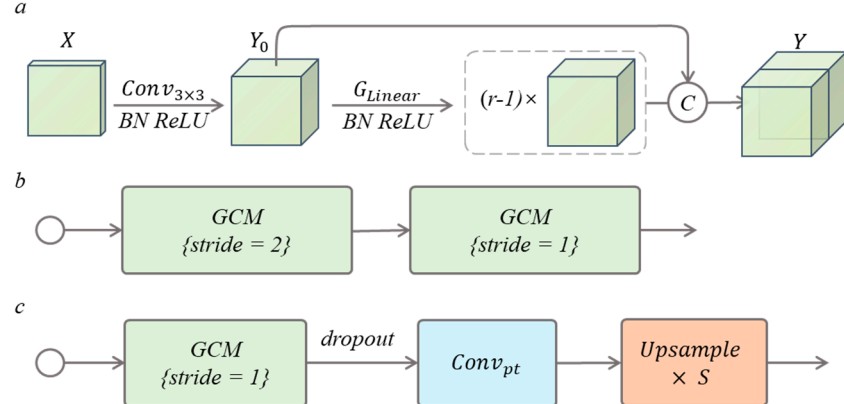

**Figure 4.** Structure illustration of (**a**) Ghost Convolution Module (GCM): initial feature maps are generated by $Conv_{3\times3}$ and expanded through $G_{linear}$ ; (**b**) Ghost Block {stride 2-1}: is combined with 2 *GCM* with stride = 2 and 1 in order; (**c**) Ghost Convolution Network Segmentation Head (GCN-Head).

As output channels are normally predefined, the initial channels are dependent on $r$, and $w' = \frac{w_o}{r}$. It is worth noting that the computation cost of $G_{linear}$ is much smaller than conventional convolution, and the ratio $r$ indicates a speed-up ratio compared to ordinary convolution, as proven in [44]. Although a large $r$ ratio may bring computation costs down, it can lead to instability for lacking cross-channel information fusion.

Taking advantage of GCM, Ghost Block (G-Block) is introduced for the Detail Branch of the bilateral segmentation network. Each G-Block starts with a GCM of stride 2 to reduce computation cost, and is followed by one or two GCM of stride 1 to deepen the feature extraction network, as shown in Figure 4b. the structure is simple yet effective for Detail Branch with a wide feature dimension. Moreover, we also noticed another computation-heavy part of BiSeNetV2 is the FCN head [45], so a Ghost Convolution

Network segmentation Head (GCN-Head) is designed, as shown in Figure 4c. The GCM of 1 stride generates abundant feature maps first, then depthwise convolution $Conv_{pt}$ compresses the channel number to the number of classes. After a dropout layer to avoid overfitting, feature maps are upsampled to the input shape by bilinear interpolation.

### 2.3.2. Ghost Bilateral Network (GBiNet)

The proposed Ghost Bilateral Network (GBiNet) is based on the BiSeNetV2 [42], and the overview of GBiNet is shown in Figure 5. There are four main parts: Detail Branch, Semantic Branch, Aggregation Layer, and Segmentation Head for inference. Moreover, a set of Auxiliary Segmentation Heads works in the training phase only.

**Table 3.** Detailed Parameters of Standard *GBiNet*.

| Stage | Input Shape | Operator | Number and Stride | Output Shape |
|---|---|---|---|---|
| Encoder—Detail Branch 1 | H × W × 3 | G-Block [1] | 2-1 | H/2 × W/2 × 64 |
| Encoder—Detail Branch 2 | H/2 × W/2 × 64 | G-Block [1] | 2-1-1 | H/4 × W/4 × 64 |
| Encoder—Detail Branch 3 | H/4 × W/4 × 64 | G-Block [1] | 2-1-1 | H/8 × W/8 × 128 |
| Encoder—Semantic Branch 1 | H × W × 3 | Stem-Block [3] | 4 | H/4 × W/4 × 16 |
| Encoder—Semantic Branch 3 | H/4 × W/4 × 16 | GE-Block [3] | 2-1 | H/8 × W/8 × 32 |
| Encoder—Semantic Branch 4 | H/8 × W/8 × 32 | GE-Block [3] | 2-1 | H/16 × W/16 × 64 |
| Encoder—Semantic Branch 5 | H/16 × W/16 × 64 | GE-Block [3] | 2-1-1-1 | H/32 × W/32 × 128 |
| Encoder—Semantic Branch 5 | H/16 × W/16 × 64 | CE-Block [3] | 1 | H/32 × W/32 × 128 |
| Encoder—Aggregation Layer | (H/8 × W/8 + H/32 × W/32) × 128 | BGA-Block [3] | 1 | H/8 × W/8 × 128 |
| Decoder—Segmentation Head | H/8 × W/8 × 128 | GCN-Head [2] | 1 | H × W × 5 |

[1] G-Block denotes Ghost Convolution Block. [2] GCN-Head means Ghost Convolution Networks for Semantic Segmentation. [3] Stem, GE, CE, and BGA-Block indicate the stem block, gather–expansion block, context embedding, and bilateral guided aggregation layer referring to BiSeNetV2.

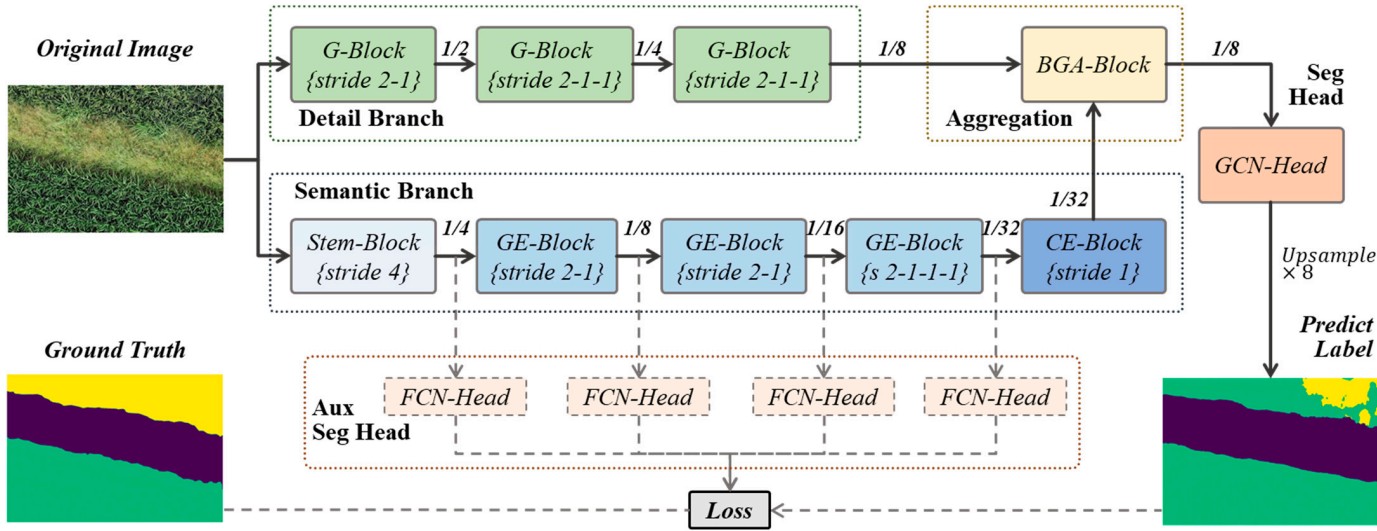

**Figure 5.** Overview of Ghost Bilateral Network (GBiNet). The network has 4 main parts: the Detail Branch, Semantic Branch, Aggregation Layer, and GCN Segmentation Head. Each box is an operation block, and the arrow connection represents the feature maps flow with numbers above showing the ratios of map size to the input size. The detail of each stage is available in Table 3.

A bilateral network has two different encoder branches. The Detail Branch requires high channel capacity to encode rich spatial details. Meanwhile, as this branch focuses only on low-level features, a shallow and wide-layer structure is preferred. On the contrary, Semantic Branch aims to capture high-level semantics, such that a deep and narrow layer structure is needed. Because of the different shapes of output feature maps from two

parallel branches, the bilateral guided aggregation (BGA) block is used to merge the two complementary features. The Auxiliary Segmentation Heads are inserted at different stages of the network, which can enhance the feature representation during the training phase while adding no computation costs in the inference stage.

According to the floating-point operations (FLOPs) of each component, the Semantic Branch and BGA Block consume less than 10% of the overall computation cost in the original BiSeNetV2 model. The fast down-sampling strategy, lightweight depthwise convolution, and global pooling embedding have been adopted in the original stem block (Stem-Block), gather-and-expansion layer (GE-Block), and context embedding block (CE-Block), respectively, in the original Semantic Branch. Additionally, the GBA block uses the context information of Semantic Branches to guide the feature response of Detail Branches. Different scale feature representations can be captured through different scale guidance, which enables effective communication between two branches. So, the components and structures of these two parts are kept in our GBiNet. For details, please refer to [42].

The other two parts, Detail Branch and Segmentation Head, are the bottlenecks of speeding up model inferencing. Therefore, we have improved both parts with GCM to construct a more efficient model. Table 3 depicts the overall architecture of standard GBiNet. The Detail Branch mainly consists of a stack of G-Block with GCM as the basic unit. Each block implies a stage according to the sizes of input feature maps. All GCM are applied with $stride = 1$, except that the first one in each stage or G-Block is with $stride = 2$. The implicit kernel channels of each stage are {64, 64, 128}, and the increase of the channels is also at the first GCM. The last output channel number of the Detail Branch is the same as that of the Semantic Branch, whose implicit kernel channels are {16, 32, 64, 128}, which indicates the bandwidth from the encoder to the decoder. The divergent strides combination of the two branches benefits multi-scale feature extraction for semantic segmentation tasks. After bilateral aggregation, a GCM extends the features to 1024 decode channels, and a pointwise convolution layer with dropout is utilized to decode feature maps to a 5-channel mask for segmentation. An interpolation layer of 8-times upsampling is also applied to yield the final mask, as pictured in Figure 4c.

We have also designed a tiny model, namely GBiNet_t. The output channels of its Detail and Semantic Branch are {16, 16, 32} and {8, 16, 16, 32}. Additionally, all G-Blocks at the Detail Branch are of strides {2, 1}, and the default number of decode channels is 32 for GBiNet_t.

### 2.3.3. Experimental Setup and Parameters

To evaluate the performance of the proposed models on the multi-stage rice field segmentation tasks, we have trained GBiNet and other existing models on the PaddySeg dataset. All models are trained from scratch with the Kaiming initialization method [46]. For all training schedules, the optimizer is stochastic gradient descent (SGD) with a 0.9 momentum and $5e^{-4}$ weight decay. The initial learning rate is $5e^{-3}$ and is decayed using a poly strategy with 0.9 power and $1e^{-4}$ minimum rate. The max training iterations is $60k$, which is abundant for all models to fully fitted on PaddySeg, and the training weights would be saved and evaluated every $6k$ iterations. The best-trained weights on the evaluation dataset were kept for testing.

For evaluation, mean intersection over union (mIoU) is the primary metric. Meanwhile, overall pixel accuracy (aAcc) and intersection over union (IoU) of each class were also recorded for balance assessment over multi-classes. The model weight's number and FLOPs were measured to evaluate the memory and computation costs.

For training data augmentation, we randomly resize, randomly crop, and randomly horizontally flip the images to a target size. The scale range of resizing is $0.5 \sim 2.0$, and 0 and 255 would pad the absent area of the image and its mask. For the test pipeline, no augmentation method is adopted. There are two kinds of data processing modes, sliding window (Slide) and down-sampling whole (DSW). The first mode has a target size of $546 \times 546$, and during inference, a $546 \times 546$ sampling window would Slide on the

$1638 \times 1092$ image with (364, 364) stride and yield 6 patches to feed the model. While DSW has an $819 \times 546$ target size, it down-samples the image to the target size and feeds the whole image to the model. The DSW input mode effectively simulates images taken from twice the height (50 m) compared with the Slide mode (25 m) and larger ground sampling distance (GSD). All random processes used the random seed of 2022 to ensure reproducibility. The model implementation code was arranged in MMSegmentation style under PaddySeg/GBiNet/directory at (https://github.com/HobbitArmy/PaddySeg, accessed on 3 January 2023).

All training progress was conducted on the server with two RTX-2080Ti GPUs, and each GPU takes 4 samples per iteration; thus, the overall batch size is 8. To test the practical inference speed, we measured frames per second (FPS) on the PC with one RTX-2070 GPU. We fed the test set (520 samples) to the pending model and recorded its processing FPS, then repeated this 5 times to eliminate fluctuation.

### *2.4. Traits Locating and Mapping System*

To achieve the fast locating and mapping of crop traits, we used this section to provide a system to process the original UAV images directly, which does not require structure-from-motion (SFM) and image mosaic. The distribution map of target traits can be obtained, combined with the feature extraction methods demonstrated in the previous section. Section 2.4.1 introduces the geo-locating method of every pixie in a UAV image based on a simplified photogrammetry model. In Section 2.4.2, we split the UAV image into patches in sequence and discard some patches to achieve the sparse geo-distribution.

#### 2.4.1. Direct Geo-Locating (DGL)

According to photogrammetry, to calculate the absolute location of the target point in an image, the coordinate transformation relationship between image points in different spatial rectangular coordinate systems should be established first [47]. This can be complicated for UAV photographs, where a lot of exterior elements are required to achieve multiple transformations. Photos are transformed from image plane to image space, then to the gimbal and the aircraft body, and finally, to a geodetic coordinate system (GCS).

Fortunately, with a three-axis gimbal, vertical photographs were taken with the optical axis of the camera vertical to the ground [48]. As Figure 6a illustrates, the roll or pitch of the aircraft would not cause deflection of the camera, and only rotational movement leads to the yaw angle changing of the camera [49]. Additionally, the distortion of the lens was calibrated, and the RTK position recorded in the image EXIF information is accurate and was calibrated from the RTK antenna to the principal point of the camera. Based on this, for a flat rice field, we simplify the imaging process to a central projection model with the subject plane *G-XY* parallel to the camera sensor plane *o-xy*, as shown in Figure 6b. Using the internal and external parameters of the camera and the collinear equation, the absolute location of the target point in a photo can be direct geo-located (DGL).

More specifically, for GCS *G-XYZ*, y-axis points to the north, x-axis points to the east, and the z-axis points upwards. For the auxiliary imaging coordinate system *o-xyz*, *o* is the central point of imaging film, the y-axis points to the top of the image, the x-axis to the right, *S* is the exposure station (also lens center) at $[X_s, Y_s, Z_s]$, *N* is the vertical ground projection point, *S-N* is the optical axis position, the length of *S-o* is the focal length, and $\kappa$ is the counterclockwise angle from the y-axis to the north (y-axis). The target point *T* is collinear with the corresponding imaging point *t* and the projection center *S*, and $\alpha$ denotes the angle from the y-axis to the vector $\overrightarrow{ot}$. The coordinate of *T* at *G-XY* plain can be calculated from the following equation:

$$coord_T = coord_N + \overrightarrow{NT} = [X_s, Y_s] + \overrightarrow{NT} \tag{3}$$

$$\overrightarrow{NT} = \frac{|SN|}{|So|} * \overrightarrow{ot} = \frac{|SN|}{|So|} * |ot| * [\sin(\kappa - \alpha), \cos(\kappa - \alpha)] \tag{4}$$

where $|SN|$ is the relative height of the exposure station, $|So|$ is the focal length, $|ot|$ is the distance between the central point and the target point on the camera sensor, which can be calculated by:

$$|ot| = pixel_{dis} * \frac{size_{sensor}}{pixel_{sensor}} \tag{5}$$

where $pixel_{dis}$ is the pixel distance between the central point and target point, $size_{sensor}$ is the effective sensor size of the camera (along the width or length), and $pxiel_{sensor}$ is the corresponding pixel number.

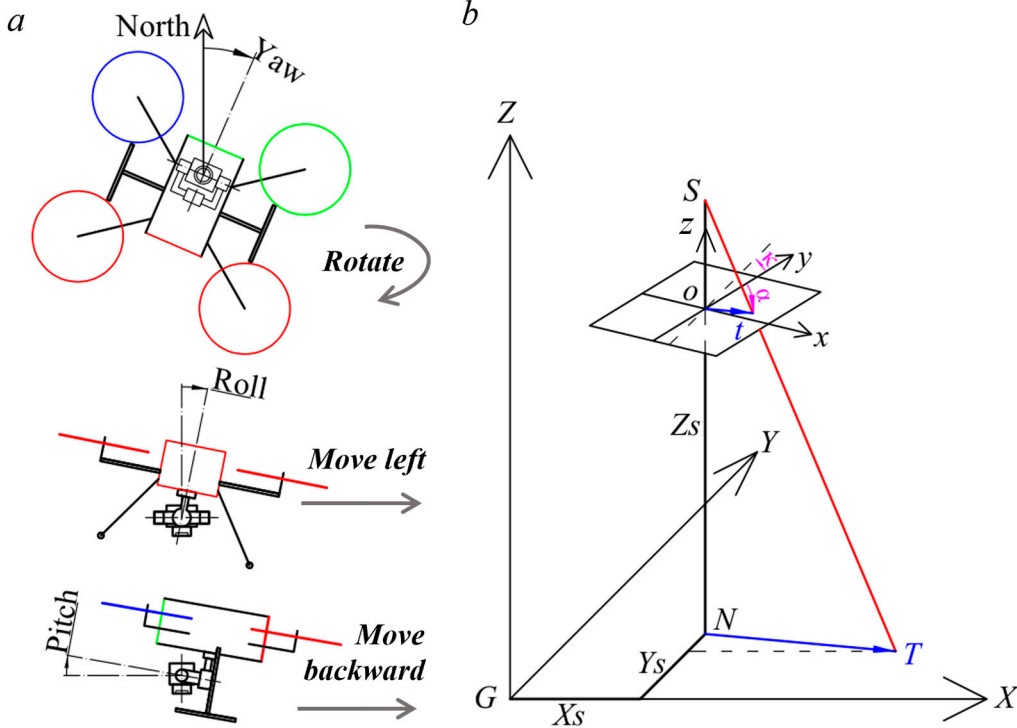

**Figure 6.** Direct Geo-Locating Based on (**a**) Exterior Orientation and (**b**) Central Projection at Nadir View.

In practical use, we have adopted the calibrated $size_{sensor}$ and focal length yielded by the Pix4D project rather than the labeled value. For our P1 camera, the average focal length at 25 m relative height is 34.90 mm, and the sensor size is 35.0000 × 23.3276 mm. These three values are fixed after initialization, and other parameters are loaded from metadata in the image. For convenience, we implemented the DGL method with Python and packed it up into class. See the code DirectGL.py for more details.

The ground control points (GCP) were set and located using a high-accuracy handheld RTK to evaluate the locating accuracy of DGL, as shown in Figure 7a,b. Then we used a UAV to scan the area and load the UAV image. In Figure 7c,d, given the measured $GCP_{loc}$, the predicted GCP location $GCP_{pred}$ on the image is plotted with a blue dot. By selecting the observed $GCP_{obs}$ manually, the haversine distance between $GCP_{pred}$ and $GCP_{obs}$ is calculated and recorded.

2.4.2. Incremental Sparse Sampling (ISS)

Due to the high overlap of UAV images, repeated calculation of the same region is unavailing and brings high latency to the system. This section supplies an incremental sparse sampling method for UAV image sequence, which splits images one by one into patches and discards the redundant ones.

As depicted in the left side of Figure 8, the whole UAV image is split into 5 rows and 5 column patches first. Secondly, geo-locations of the central 3 × 3 candidate patches are calculated (considering the DGL-acc, the outer edge patches of each image are removed).

Patches of the first image in sequence would be accepted directly and stored. The distance matrix between the subsequent patches set of one image and the stored patches is calculated. Any candidate patch that is too close to the stored patches ($< dis_{patch} * r_{elm}$) would be excluded, where $dis_{patch}$ is the short distance between two patches in one image, and $r_{elm}$ is the distance ratio.

Because the stored patches are dynamically changing, and patches to be stored are selected step by step, we call this method incremental sparse sampling (ISS). ISS method is summarized in Algorithm A1 of Appendix A and implemented in the PatchSparseSampling.py code file.

As shown on the right side of Figure 8, after sparse sampling, the label mask is computed for each stored patch using a segmentation model. Moreover, the mask is averaged pooled to down-sampling the trait's value, which is like splitting the patch into smaller boxes, and the pooling kernel size is the box size. This operation reduces data redundancy while improving robustness. The image location of every box is recalled to DGL their geo-location (see code Patch2Distribution.py for detail). With the traits value-location pairs, the rice phenology distribution map is acquired using inverse distance weighted (IDW) interpolation.

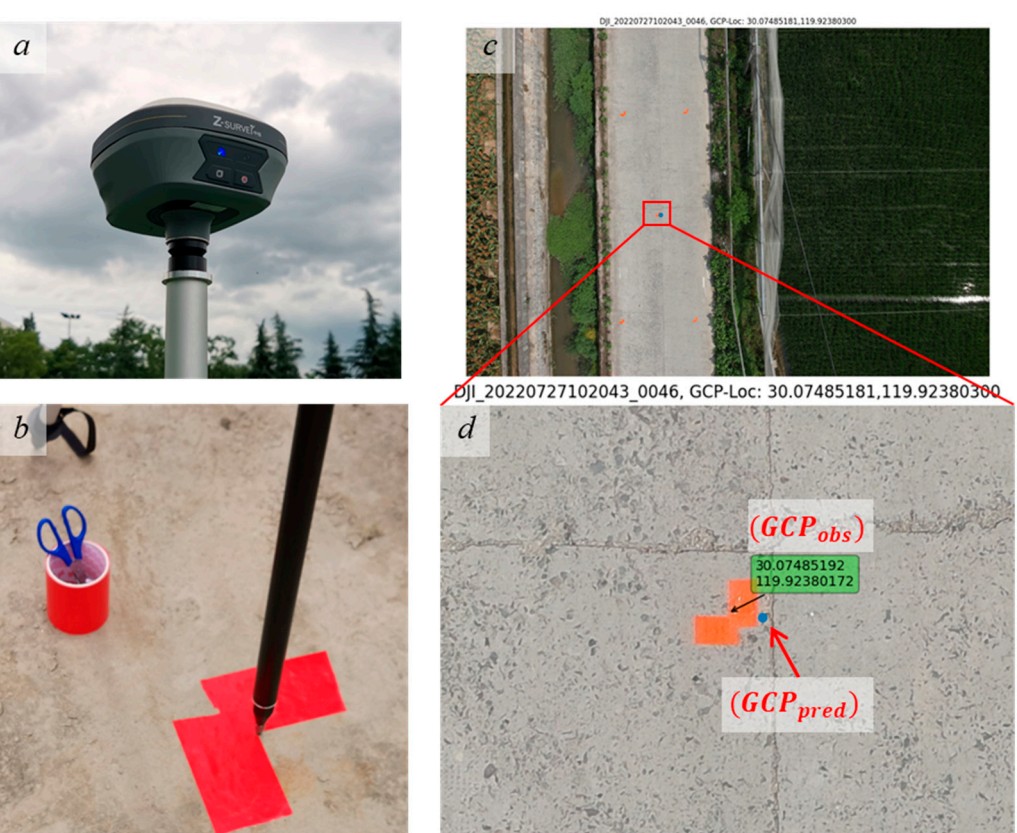

**Figure 7.** Evaluation of Direct Geo-Locating with (**a**) RTK and (**b**) Ground Control Points. Measuring the distance between the observed GCP location and the predicted location (marked with a blue dot) in one UAV image (**c**): Original Image; (**d**): Detail View of GCP.

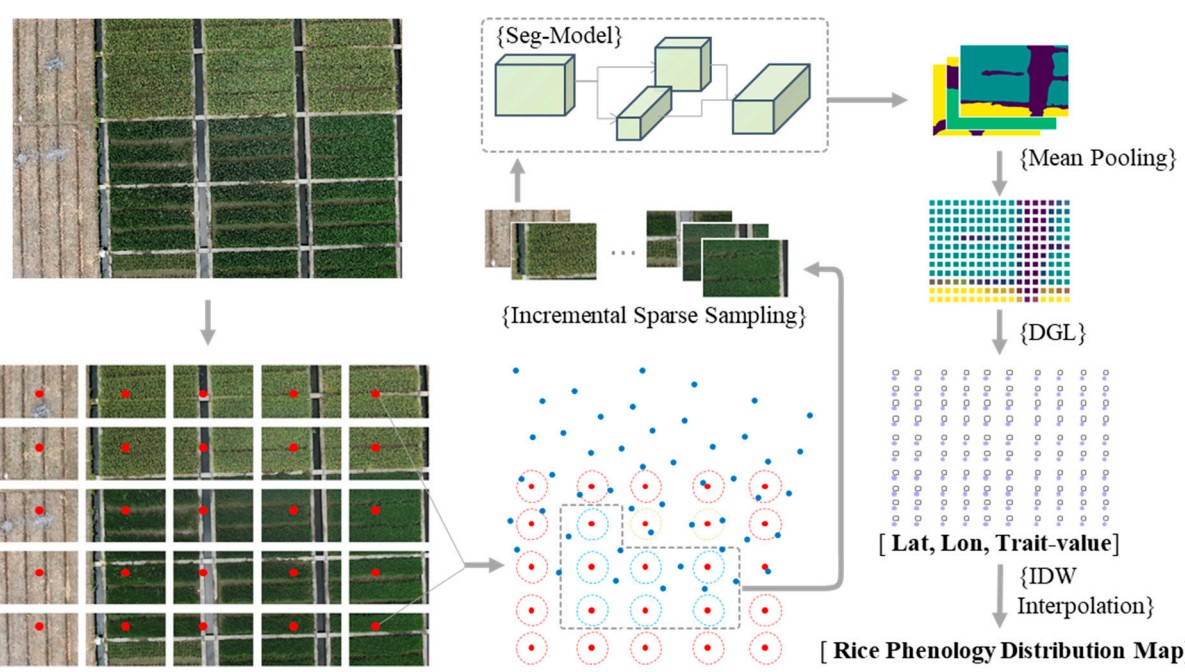

**Figure 8.** Incremental Sparse Sampling and System Workflow.

## 3. Results

In this part, we first report the performances of GBiNet and current classic semantic segmentation models on the PaddySeg dataset in Section 3.1. Then, in Section 3.2, the accuracy of direct geo-locating is tested. Finally, Section 3.3 shows a practical test of rice phenology distribution mapping that integrates all methods for practical use.

### 3.1. Segmentation Model Performance

According to the models, training methods, and hyper-parameters described in Section 2.3.3, a series of training experiments were conducted. Four classical semantic segmentation networks were trained and evaluated on PaddySeg, as summarized in Table 4. Apart from the segmentation accuracy indicators mean intersection over union (mIoU) on validation and test set, the costs of processing on memory, computation, and time are reflected by weights number (in Million units), floating-point operations (FLOPs, in Giga unit) and frame per second (FPS), respectively. Towards real-time aerial image processing, speed and accuracy are the key parameters. Apart from the sliding-window (Slide) input mode, an extra experiment of the down-sampling whole (DSW) input mode was conducted.

**Table 4.** The Performance of Existing Models on PaddySeg (Slide Input: 546 × 546, Speed Tests on RTX2070).

| Model | Weights (M) | FLOPs (G) | Speed (FPS) | mIoU-val % | aAcc-Test % | mIoU-Test % | DSW Speed (FPS) |
|---|---|---|---|---|---|---|---|
| pspnet_r18-d8 (2017) | 12.79 | 62.98 | 2.8 | 91.78 | 95.53 | 91.57 | 21.3 |
| deeplabv3p_r18-d8 (2018) | 12.47 | 62.91 | 2.7 | 92.16 | 95.56 | 91.55 | 20.7 |
| fcn_hr18s (2019) | 3.94 | 11.13 | 2.5 | 91.82 | 95.39 | 91.28 | 21.7 |
| bisenetv2_fcn (2021) | 14.77 | 14.25 | 5.4 | 91.52 | 95.41 | 91.31 | 36.4 |

At Slide input mode, all models achieved a mIoU on the test set above 91%, with the highest pspnet reaching 91.57%. At the same time, the inference speed of BiSeNetV2 (5.4 FPS) exceeds other models with an average of 2.7 FPS. However, this level of latency is still too high for practical use. Fortunately, all models acquired a six- to nine-times boosting speed from DSW input mode, as shown in the last column of Table 4. Moreover, the price

is a minor performance reduction according to the 0.36% test set mIoU drop of BiSeNetV2 with DSW mode, as revealed in Table 5. Therefore, in the following experiment, we only adopted the DSW input mode.

**Table 5.** The Performance of GBiNet on PaddySeg (DSW input: 819 × 546, Speed Tests on RTX2070).

| Model | Weights (M) | FLOPs (G) | DSW Speed (FPS) | mIoU-val | aAcc-Test | mIoU-Test |
|---|---|---|---|---|---|---|
| bisenetv2_fcn | 14.77 | 21.29 | 36.4 | 91.25 | 95.11 | 90.95 |
| GBiNet_r2 | 13.93 | 12.22 | 41.0 | 91.64 | 95.43 | 91.50 |
| GBiNet_r4 | 13.51 | 7.68 | 44.9 | 91.13 | 94.89 | 90.47 |
| GBiNet_r8 | 13.30 | 5.41 | 46.8 | 90.92 | 94.93 | 90.56 |
| GBiNet_64dx4_r2 | 3.51 | 3.03 | 47.9 | 90.74 | 94.79 | 90.26 |
| GBiNet_64dx8_r4 | 3.34 | 2.24 | 52.3 | 90.90 | 94.80 | 90.40 |
| GBiNet_t32dx2_r4 | 0.82 | 0.50 | 61.9 | 90.20 | 94.71 | 90.19 |

The red and cyan lines in Figure 9a show the convergence process of the BiSeNetV2 model with Slide and DSW input mode. DSW mode leads to a much lower loss value for training, while their average accuracies on the validation set are almost the same (Figure 9b). This implies that coarse images are more easily segmented by the model, resulting in lower training loss values.

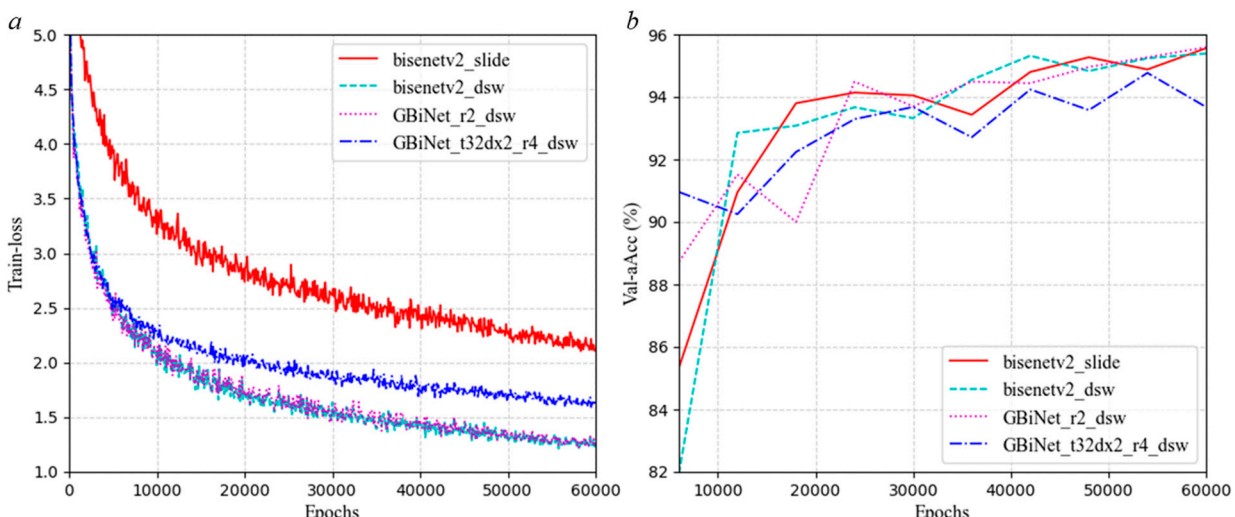

**Figure 9.** Convergence process of the models: (**a**). Loss descending during training; (**b**). Average segmentation accuracy on the validation set.

To explore the potential of the proposed method comprehensively, we designed multiple GBiNet and its tiny sibling GBiNet_t. As Table 5 depicts, the number after _r suffix denotes the ghost ratio, the number right after *GBiNet_* or *GBiNet_t* is the last output channels (LOC) of the Detail and Semantic Branches for tuning the width of model, and the number after *dx* is the expanding ratio of decode channels in GCN-Head. Accordingly, if the LOC number is 64, then the output channels of its Detail and Semantic Branches are {32, 32, 64} and {16, 32, 32, 64}; if the number is 32, then the output channels are {16, 16, 32} and {8, 16, 16, 32}. If not specified, all parameters are set as detailed in Section 2.3.2 by default. For example, GBiNet_r2 has {64, 64, 128} and {16, 32, 64, 128} output channels of the two branches, 1024 decode channels, and the ghost ratio of two, while GBiNet_t32dx2_r4 has {16, 16, 32} and {8, 16, 16, 32} output channels, 64 decode channels, and ghost ratio of four.

From the results in Table 5, the speed of the proposed models surpassed all existing models. When the model ghost ratio increased from two to eight, the FLOPs reduced significantly, and the speed rose accordingly, while the accuracy decreased as expected.

In particular, the GBiNet_r2 performed even slightly better than the original BiSeNetV2, achieving the highest 91.5% mIoU-test performance and lower FLOPs. According to Figure 9, the convergence curves of GBiNet_r2 and BiSeNetV2 almost coincide, verifying the robustness of the ghost convolution. Furthermore, as we changed the LOC and decode channel number smaller, the speed of the model increased significantly, boosting the tiny model GBiNet_t32dx2_r4 to a speed approaching 62 FPS with an acceptable mIoU around 90.2% on the test set.

The ground truth and segmentation results of the most accurate and fast GBiNet model are shown in Figure 10. Some spots in the ground truth mask of the last column are weeds among rice plants which are hard to be identified by models.

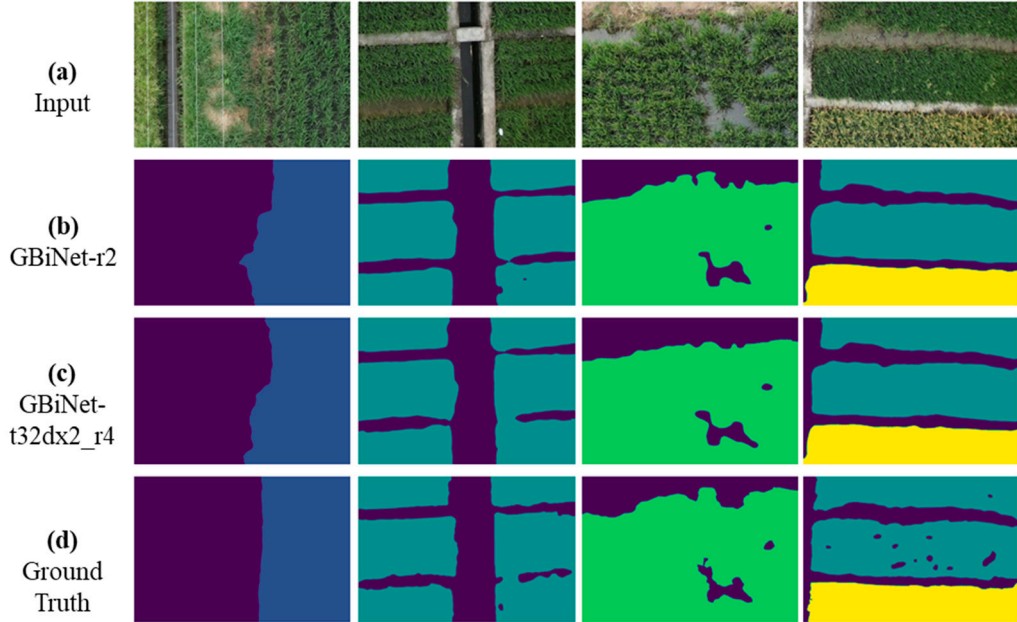

**Figure 10.** Segmentation Result of Rice Field at Different Stages.

*3.2. Direct Geo-Locating Accuracy*

Although direct geo-locating (DGL) is easy to achieve in principle, this section gives the result of actual locating experiments to verify its feasibility. Many aspects of reasons can lead to the deviation of final positioning, including inaccuracy of the airborne RTK system, instability of the gimbal, and camera distortion. To assess the DGL system comprehensively, we directly calculated the haversine distance of the true location and predicted location as the deviation.

A total of 20 ground control points (GCP) have been set at three sites with their accurately measured geographic coordinate locations (GCP_loc) using handheld RTK, as shown in Table A1 of Appendix A. Each GCP (with a GCP_name) would be observed by several UAV images (counted as OBS_num), and the distance between GCP_loc measured on the ground and the observed location estimated on the aerial image was recorded. We average all distance deviations of each GCP (DEV_avg) and found that the locating deviation of an arbitrary point is near 0.21 m.

The distributions of all observed deviations are fitted with communally used functions in Figure 11. Moreover, the expectations of these functions are also around 0.2 m, which is less than 1% of the relative flying height. This accuracy level, coupled with smoothing filtering of interpolation during the subsequent mapping process, is sufficient for agricultural field usage.

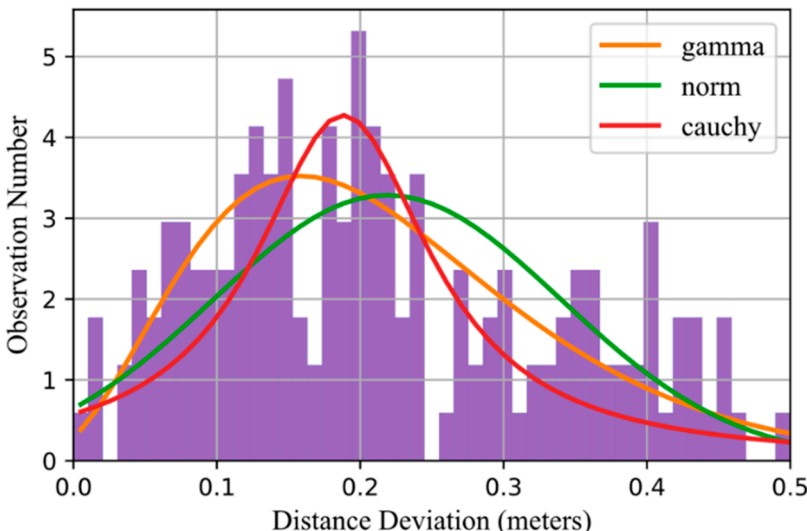

**Figure 11.** GSD Distance Deviation Distribution.

### 3.3. Rice Phenology Mapping

After feature extraction and trait locating, an integrated test on rice phenology distribution mapping is presented. Figure 12 depicts the key intermediate diagram generated in the system workflow as described in Section 2.4.2. Each blue star in Figure 12a is a waypoint where an image is taken by the UAV with a red sequence number marked aside. In Figure 12b, these image geo-locations (IGL) are marked as blue stars as well, and the red points are sparsely sampled patch geo-location (PGL). Because the box geo-locations (BGL) in one patch are arrayed one by one, they are displayed like a purple block around the PGL point.

PGL points distribute evenly thanks to the sparse sampling, and the total patch number to be processed further is decreased from 1170 (130 images * nine patches each) to 294. This greatly saved the computation costs. Although most of the target field is covered by purple, some of the regions are ignored mistakenly, as shown blank in Figure 12b.

Based on the BGL locations and corresponding trait values (rice phenology of the box area), we used inverse distance weighted (IDW) interpolation in QGIS software and generated a heat map of rice phenology distribution, as Figure 12c shows. The color mask represents others, seedling, jointing, heading, and filling from blue to green to red, with values exchanging from 0 to 4. Areas marked with white numbers from 1 to 4 are rice. We hid the zero value in the figure so that roads, bare soil, trees, and other regions are blank. Moreover, the rice area is also represented by different colors and values according to the phenology stage.

The rice growing period in different plots can be judged with this picture. For example, the field to the north and the roadside strip areas are mainly at tillering stage. While rice planting troughs in the middle are mainly at the filling stage, the east side troughs are mainly at the jointing stage.

In Figure 12c, there is an unusual blue area in the center of the planting troughs area. As we zoom in, as shown in Figure 13a, it is obvious that the low-value region (circled in white) is caused by the presence of weeds. Because of the fine label of PaddySeg, weeds amongst rice fields are also considered to be others, so the model judges this area as non-rice. Another area circled in yellow at the bottom right is the rice area, but it lacks a number mark, which is caused by the incomplete patch coverage mentioned earlier. However, the existence of interpolation makes the final mapping results more robust, where the color of this area still indicates that it is rice at the filling stage (Figure 13b).

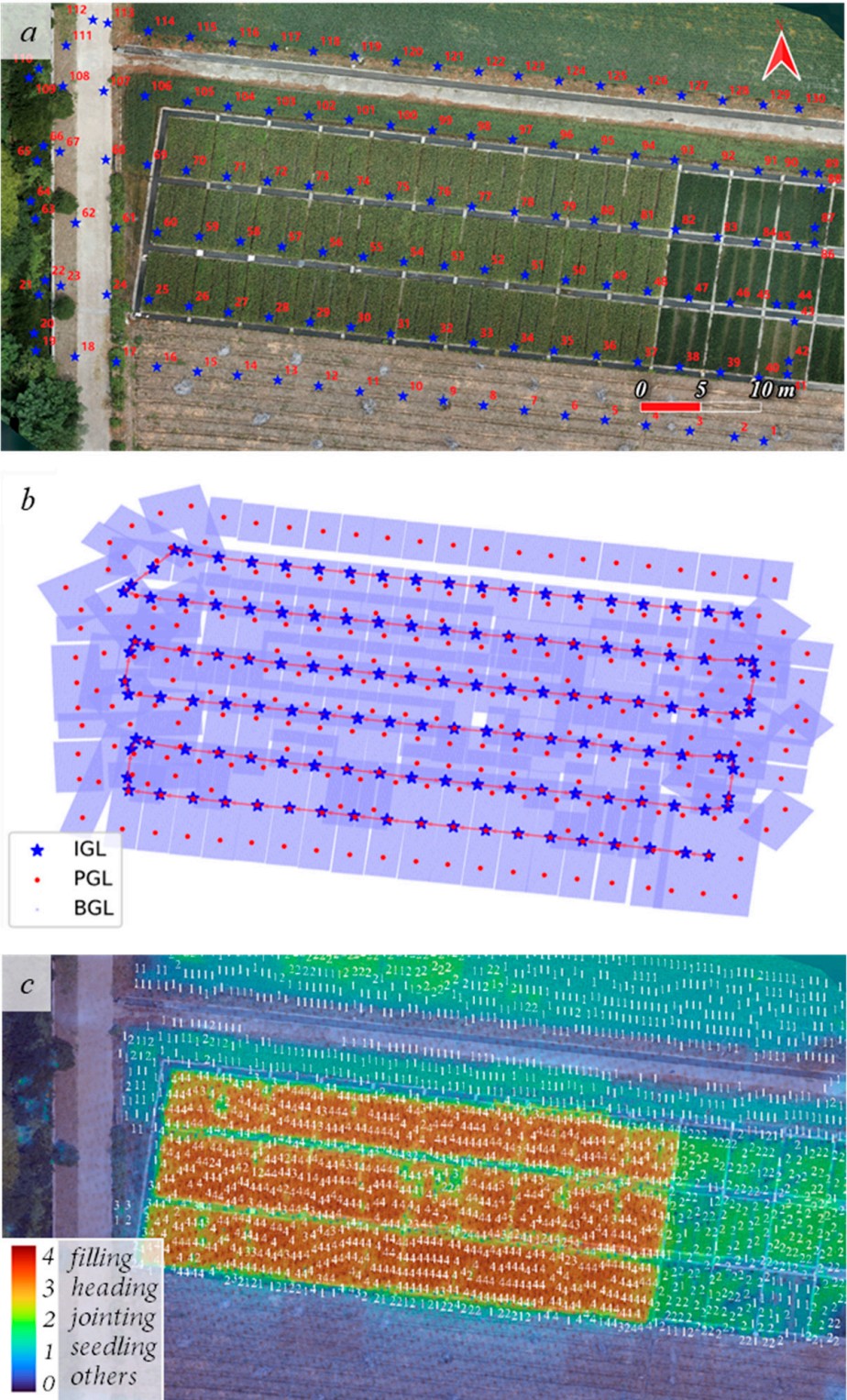

**Figure 12.** Rice Phenology Mapping at DJD-5 Experimental Field: (**a**). Drone Waypoints of Image Capture; (**b**). Sparse Sampled Patch-Boxes Distribution; (**c**). Distribution Map of Rice Phenology with Interpolation.

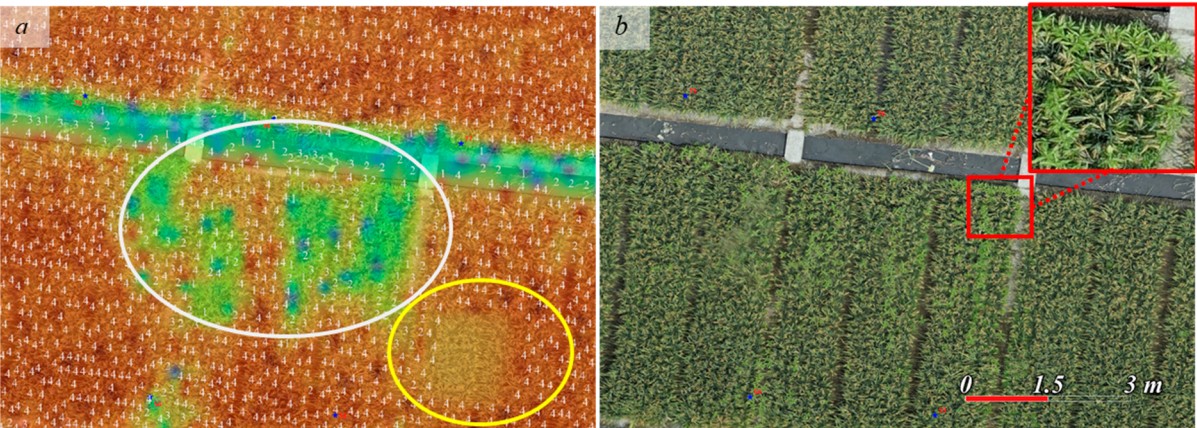

**Figure 13.** Detail View of the Singular Region in Phenology Map.

## 4. Discussion

### 4.1. Efficiency of GBiNet

Bisenetv2 is an efficient real-time semantic segmentation network that employs a bilateral structure in the decoder portion. However, the calculation cost of the detailed branch and decoder portion is too high, accounting for nearly 90%, as shown in Table 6. Therefore, this paper proposed a ghost convolution module to improve and reduce the computation consumption of these two parts. By adopting a ghost ratio of two alone, half of the computation was reduced, and the computation continued to decrease as the ratio increased. In addition, a tiny model called GBiNet_t was also proposed, which compressed the overall computation to less than 0.5 GFLOPs. The tiny model not only reduced the computation but also cut down the number of parameters. Moreover, the smallest model has only 0.091M parameters, which has the potential to provide real-time computation and be deployed on an edge device.

**Table 6.** Computation Costs and Parameters Number of GBiNet and BiSeNetV2.

| Computation/ Parameters | Model | Overall | Components | | | | |
|---|---|---|---|---|---|---|---|
| | | | Detail Branch | Semantic Branch | BGA Layer | Decode Head | Auxiliary Head |
| FLOPs (G) | bisenetv2_fcn | 21.288 | 10.113 | 1.223 | 1.525 | 8.427 | 0 |
| | GBiNet_r2 | 12.222 | 5.191 | 1.223 | 1.525 | 4.283 | 0 |
| | GBiNet_r4 | 7.681 | 2.730 | 1.223 | 1.525 | 2.203 | 0 |
| | GBiNet_r8 | 5.412 | 1.500 | 1.223 | 1.525 | 1.164 | 0 |
| | GBiNet_64dx8_r4 | 2.443 | 0.749 | 0.731 | 0.385 | 0.578 | 0 |
| | GBiNet_t32dx2_r4 | 0.499 | 0.182 | 0.180 | 0.098 | 0.039 | 0 |
| Weights (M) | bisenetv2_fcn | 3.343 | 0.519 | 1.160 | 0.479 | 1.185 | 11.421 |
| | GBiNet_r2 | 2.504 | 0.263 | 1.160 | 0.479 | 0.602 | 11.421 |
| | GBiNet_r4 | 2.084 | 0.136 | 1.160 | 0.479 | 0.309 | 11.421 |
| | GBiNet_r8 | 1.874 | 0.072 | 1.160 | 0.479 | 0.163 | 11.421 |
| | GBiNet_64dx8_r4 | 0.577 | 0.036 | 0.339 | 0.121 | 0.081 | 2.902 |
| | GBiNet_t32dx2_r4 | 0.091 | 0.006 | 0.049 | 0.031 | 0.005 | 0.726 |

It is not difficult to reduce the network size; the key is how to maintain satisfactory accuracy with simultaneous fast processing. Ghost convolution uses grouped pointwise linear transformations, where group convolution can be seen as a decoupling of the original convolution operation, improving the sparsity between filters in the original convolution operation and playing a regularization role to some extent [50]. This is one of the reasons why GBiNet_r2 performs better than BiSeNetv2.

### 4.2. Confusion Matrix and Classes Accuracy

As previously mentioned, the number of sample pixels was unbalanced between different categories. The majority class may acquire biased attention from the model during training and reach a higher recall ratio, but this is not the case in our experiments. As shown in Table 7, although the "others" class had the largest number of samples, its overall classification accuracy was the lowest. While the "heading" and "seedling" classes had a smaller number of samples, their accuracy was also at a moderate level.

**Table 7.** Segmentation Results over Multi-Classes on PaddySeg Test Set.

| Model | Class Pixel Number Ratio and IoU-Class (%) | | | | |
|---|---|---|---|---|---|
| | Seedling 13% | Jointing 24% | Heading 13% | Filling 18% | Others 32% |
| bisenetv2_fcn | 91.79 | 24% | 88.20 | 94.38 | 87.54 |
| GBiNet_r2 | 91.62 | 93.43 | 89.85 | 94.31 | 88.29 |
| GBiNet_r4 | 91.29 | 92.21 | 87.18 | 94.21 | 87.48 |
| GBiNet_r8 | 90.49 | 92.59 | 88.67 | 93.58 | 87.46 |
| GBiNet_64dx8_r4 | 89.98 | 91.90 | 89.39 | 93.74 | 87.01 |
| GBiNet_64dx4_r2 | 90.42 | 91.82 | 87.72 | 93.89 | 87.45 |
| GBiNet_t32dx2_r4 | 88.83 | 92.31 | 89.90 | 93.31 | 86.61 |
| Average | 90.63 | 79.21 | 88.70 | 93.92 | 87.41 |

The minority was not ignored by the model, while the majority class gained no bonus from its number as well. Setting class weights to weaken the majority class of "others" and enhance the minority class of "heading" or "seedling" would exacerbate the imbalance. Therefore, in this study, class weights were not used in the loss function.

The confusion matrix of the GBiNet_r2 model segmentation class accuracy on the test set is shown in Figure 14. Most of the misclassifications are caused by predicting various stages of rice as "others", with an error rate of around 2% to 4%. One reason is that the weeds amongst rice plants are also labeled as others, which are relatively difficult to identify, even for humans. Another reason is that the areas of rice at various stages are similar. Perhaps when counting the number of categories, all rice areas should have been considered as one category. In this case, others become the minority class, and the model is more biased towards jointing and filling, which is consistent with the results. It is worth exploring whether this kind of class-sample statistics can be used to equalize the model results in future experiments.

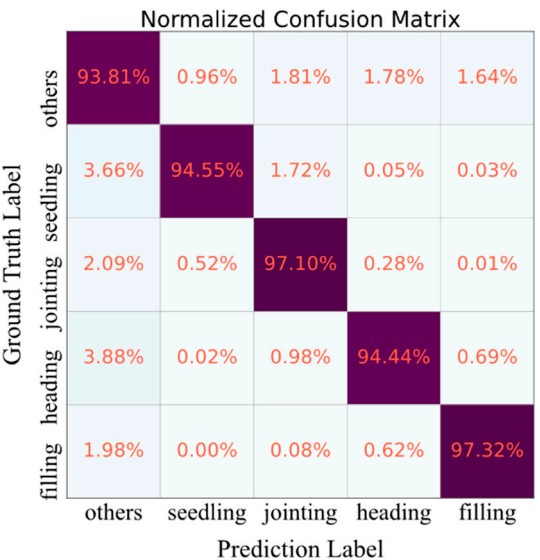

**Figure 14.** Normalized Confusion Matrix on Multi-Classes.

Additionally, the misclassification rate between adjacent categories is also relatively high, particularly when the seedling category is misclassified as the jointing category or the heading is misclassified as jointing. This result was expected, as the features of adjacent stages of rice are relatively similar and difficult to be identified.

### 4.3. Limitation and Future Study

We have proposed a range of methods and approaches for investigating rice phenological stages using UAV images. However, there are still several drawbacks: Firstly, the rice phenological stage in the PaddySeg dataset was not finely divided, only encompassing four key periods. Secondly, to ensure a clear view of key features during labeling, we chose relatively low-flight altitude UAV images, which were unsuitable for monitoring large areas. Thirdly, although the data annotation was semi-automatic, it was still time- and labor-intensive. In addition, the issue of imbalanced inter-class accuracy required further investigation and experimentation.

For future research, the direct processing workflow of UAV image sequences could be expanded to other application scenarios, including planting and harvesting progress investigations. Meanwhile, data could be collected at a higher altitude for high-efficiency large-area surveys. In an aspect of the application, we plan to seek more efficient and resolution-acceptable flight settings to facilitate the large-scale monitoring of crop traits [51], explore the feasibility of using super-resolution methods [52], enhance UAV images and downstream models, and unify UAV images at multi-levels of ground sampling ratio. For locating and mapping, we will focus on the local SFM or satellite map matching [53] methods to assist in the photogrammetric positioning of target points in a single UAV image. In terms of modeling, we will attempt to introduce super-pixel methods [54] for partitioning images first and combine this prior to processing with deep learning classification models, which will further improve the speed rather than segmentation models.

### 5. Conclusions

In this paper, to achieve automatic field rice phenology stage investigation, we have built a multi-stage rice field UAV image segmentation dataset, PaddySeg, constructed a Ghost Bilateral Network (GBiNet) for rice field segmentation and phenology classification, and designed an efficient workflow for trait locating and mapping. According to the results on PaddySeg, the most accurate GBiNet_r2 with 91.50% mIoU-test and 41 FPS speed exceeded the baseline model BiSeNetV2 (90.95%, 36 FPS). While the fastest GBiNet_t32dx2_r4 reached over 1.7 times of speed boosting (62 FPS) and kept a similar level of mIoU-test. Additionally, a straightforward and effective workflow was designed, where patches were split from the image incrementally and sampled sparsely (ISS) to eliminate computational redundancy. These patches were fed into the segmentation model that generates traits mask. Based on UAV photogrammetry, each pixel could be direct geo-located (DGL). The locating accuracy was measured on 20 ground control points (GCP) with a 21 cm average deviation (<1% relative height). The final rice phenology mapping was achieved in QGIS with an interpolation of the trait value–location pairs. This phenology mapping system could aid decision-making toward the automatic management of the rice field.

**Author Contributions:** Conceptualization, F.L. and X.L.; methodology, F.L., X.L. and J.Z.; software, X.L. and R.Y.; validation, F.L., X.L. and J.Z.; formal analysis, X.L., J.Z. and R.Y.; investigation, X.L., Y.L. and R.Y.; resources, F.L., Z.Y. and J.J.; data curation, X.L., Y.L. and J.J.; writing—original draft preparation, X.L., J.Z. and F.L.; writing—review and editing, F.L., X.L. and J.Z.; visualization, X.L., Y.L., J.Z. and R.Y.; supervision, F.L., J.J. and J.Z.; project administration, F.L., Z.Y. and J.Z.; funding acquisition, F.L. and Z.Y. All authors have read and agreed to the published version of the manuscript.

**Funding:** This research was funded by the Science and Technology Department of Guangdong Province (grant number 2019B020216001), the Science and Technology Department of Zhejiang Province (grant number 2020C02016), and the Science and Technology Department of Shenzhen (grant number CJGJZD20210408092401004).

**Data Availability Statement:** The datasets collected and generated in this study are available upon request to the corresponding author.

**Conflicts of Interest:** The authors declare no conflict of interest.

## Appendix A

---

**Algorithm A1**. Incremental Sparse Sampling of UAV Image Patches

---

**Input**: $\{I^{N \times 1}, r_p, c_p, r_{elm}, e\}$ # $I^N$ is $N$-length image index; $r_p$ and $c_p$ is the row and column number of patches set; $r_{elm}$ is the ratio that defines the minimum threshold distance, and $e$ is the edge number of patches to be removed

**Output**: $GL_l$

| | |
|---|---|
| 1: | $GL_l = [\,]$ |
| 2: | **for** $i$ in $\mathbb{Z}^{e,\cdots,r_p-e}$ **do for** $j$ in $\mathbb{Z}^{e,\cdots,c_p-e}$ **do** |
| 3: | **add** $DGL_p\,(I_1, i, j)$ **to** $GL_l$ # $DGL_p$ is direct geo-locating of a patch in the image |
| 4: | **for** $k$ in $\mathbb{Z}^{2,\cdots,N}$ **do** |
| 5: | $GL_k = [\,]$ |
| 6: | **for** $i$ in $\mathbb{Z}^{e,\cdots,r_p-e}$ **do for** j in $\mathbb{Z}^{e,\cdots,c_p-e}$ **do** |
| 7: | **add** $DGL_p\,(I_k, i, j)$ **to** $GL_k$ |
| 8: | **end for** |
| 9: | $dis_{mat}^{n \times m} = CDM\,(GL_k^n, GL_l^m)$ # $CDM$ is calculating distance-matrix |
| 10: | **for** $i$ in $\mathbb{Z}^{1,\cdots,n}$ **do for** $j$ in $\mathbb{Z}^{1,\cdots,m}$ **do** |
| 11: | **if** $dis_{mat_{i,j}} > dis_{patch} * r_{elm}$ **then** |
| 12: | **append** $GL_{k_{i,j}}$ **to** $GL_l$ |
| 13: | **end if** |
| 14: | **end for** |
| 15: | **end for** |
| 16: | **return** $GL_l$ |

**Table A1.** Direct Geo-Locating Accuracy at 3 Experimental Sites.

| Index | GCP_name | GCP_loc | OBS_num | DEV_avg (m) |
|---|---|---|---|---|
| 1 | lq1_1 | 28.21048270,121.04907560 | 7 | 0.22 |
| 2 | lq1_2 | 28.21044813,121.04889650 | 7 | 0.23 |
| 3 | lq1_3 | 28.21040046,121.04868980 | 8 | 0.24 |
| 4 | lq1_4 | 28.21034937,121.04865530 | 4 | 0.14 |
| 5 | lq1_5 | 28.21036053,121.04851120 | 8 | 0.20 |
| 6 | lq2_1 | 28.24022942,121.02827295 | 4 | 0.21 |
| 7 | lq2_2 | 28.24042392,121.02825991 | 4 | 0.19 |
| 8 | lq2_3 | 28.24053005,121.02825080 | 8 | 0.25 |
| 9 | lq2_4 | 28.24060837,121.02824626 | 8 | 0.20 |
| 10 | lq2_5 | 28.24065780,121.02824478 | 8 | 0.21 |
| 11 | sds_1 | 30.07501281,119.92426521 | 7 | 0.32 |
| 12 | sds_2 | 30.07498620,119.92427318 | 7 | 0.30 |
| 13 | sds_3 | 30.07500016,119.92423888 | 13 | 0.24 |
| 14 | sds_4 | 30.07501029,119.92420711 | 12 | 0.23 |
| 15 | sds_5 | 30.07498435,119.92420728 | 13 | 0.25 |
| 16 | sds_6 | 30.07480693,119.92382305 | 12 | 0.26 |
| 17 | sds_7 | 30.07480502,119.92378671 | 8 | 0.16 |
| 18 | sds_8 | 30.07485181,119.92380298 | 8 | 0.16 |
| 19 | sds_9 | 30.07489651,119.92381531 | 12 | 0.18 |
| 20 | sds_10 | 30.07489462,119.92378427 | 8 | 0.13 |
| | | | Sum: 166 | Avg: 0.21 |

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
