# Peer review of "Automated Rice Phenology Stage Mapping Using UAV Images and Deep Learning"

_drones, doi:10.3390/drones7020083_

Round 1

Reviewer 1 Report

The paper proposes an automatic rice phenology stage mapping using deep learning model applied to a sequence of single unmanned aerial vehicle (UAV) images. The paper is well structured with organized related works and clear figures. The experimental validation of the propose GBiNet model on the PaddySeg test-set (created by the authors containing 4 growth-stages and 2600 images) has given a Mean IoU of 91.5% which outperform the baseline model. However, several issues should be addressed:

- More comparisons have to be conducted with other models evaluated on the same test dataset to validate the efficiency of the propose method.

- It may be more efficient to work on large surfaces to accurately monitor the rice phenology stages.

- Please revise the English writing of the paper

Reviewer 2 Report

In this study, the authors presented a novel approach for extracting and locating phenological traits directly from the unmanned aerial vehicle (UAV) photograph sequence. According to results on PaddySeg test-set, the proposed GBiNet with 91.50% Mean IoU and 41 FPS outperforms the baseline model (90.95%, 36 FPS), while the fastest GBiNet_t reached 62 FPS which is 1.7 times faster than the baseline. The proposed approach demonstrated great potential towards automatic rice phenology stage surveying and mapping. The paper is well organized, but the following comments must be carefully revised to improve the quality of the paper.

1. Page2-line 88. Data collection settings should be reported in detail.

2. Page6-line 202. The model in Fig.4 lacks a lot of details, such as the number and size of convolution kernels, the size of feature maps, and so on. The authors should supplement the details of the deep learning model as much as possible to help understand its structure. The reasons for setting some hyper-parameters should be introduced, such as stride=1 or 2.

3. Page 7-line210. The structure of Figure 5 is the same problem, lacking a lot of details.

4. In Section 2.3, the authors should emphasize the reasons and motivations for adopting such a deep learning model structure in Fig. 4 and Fig. 5. How are the numbers of G-block and GE-block determined?

5. Page 11-line 372. At the beginning of the experiment, the training super parameters of the neural network model and the convergence process of the objective function should be presented.

6. In addition to its own ablation experiments, some comparative experiments with other methods should be supplemented.

7. Deep learning has demonstrated superior performance in the field of pattern classification. The following related work must be cited in Introduction or Section 1, including

“Faster mean-shift: GPU-accelerated clustering for cosine embedding-based cell segmentation and tracking.” Medical Image Analysis 71 (2021): 102048.

“Compound figure separation of biomedical images with side loss,” Deep Generative Models, and Data Augmentation, Labelling, and Imperfections. Springer, Cham, 2021. 173-183.

“Pseudo RGB-D Face Recognition,” in IEEE Sensors Journal, vol. 22, no. 22, pp. 21780-21794, 15 Nov.15, 2022.

“Improvement of generalization ability of deep CNN via implicit regularization in two-stage training process,” IEEE Access, vol. 6, pp. 15844-15869, 2018.

Reviewer 3 Report

This study uses drone imagery and deep learning to automatically map rice growth stages, and it is an interesting study. However, there are some suggestions for the author's reference to improve the rigor of the article.

1. The return method of obtaining images should be clearly explained, and the overlapping ratio of images should be provided.

2. It is mentioned in the literature review that "satellite images cannot be transmitted immediately", is this study transmitted immediately?

3. The size and error of the detection area should be provided in the text.

4. It is recommended to add a comparison using IR images.

Suggestion:

Using UAV to Detect Solar Module Fault Conditions of a Solar Power Farm with IR and Visual Image Analysis

Using Drones for Thermal Imaging Photography and Building 3D Images to Analyze the Defects of Solar Modules

5. The details of different plants should be clearly defined, such as rice=1, weeds=2, soil=3, and water=4, to improve readability.

6. It is recommended to add images of different periods to find out the failure (withered) area. And verify the failure area.

Round 2

Reviewer 2 Report

Accept.

Reviewer 3 Report

Although some limitations cannot be revised in time, hopefully, they can be addressed in future studies. Most of the proposals have been revised and accepted for publication in their current form.